# Strength in Diversity: Understanding the impacts of diverse training sets for self-supervised pre-training in histology

**Kristina L. Kupferschmidt**[1,2]                         KUPFERSK@UOGUELPH.CA
**Eu Wern Teh**[1,2]                                       ETEH@UOGUELPH.CA
**Graham W. Taylor**[1,2]                                  GWTAYLOR@UOGUELPH.CA
[1] *School of Engineering, University of Guelph, Guelph, Canada*
[2] *Vector Institute, Toronto, Canada*

## Abstract

Self-supervised learning (SSL) has demonstrated success in computer vision tasks for natural images, and recently histopathological images, where there is limited availability of annotations. Despite this, there has been limited research into how the diversity of source data used for SSL tasks impacts performance. This study quantifies changes to downstream classification of metastatic tissue in lymph node sections of the PatchCamelyon dataset when datasets from different domains (natural images, textures, histology) are used for SSL pre-training. We show that for cases with limited training data, using diverse datasets from different domains for SSL pre-training can achieve comparable performance when compared to SSL pre-training on the target dataset.

**Keywords:** digital histopathology, self-supervised learning, few-shot learning

## 1. Introduction and Background

The expensive nature of manually annotating digital histopathology images makes self-supervised learning (SSL) an appealing choice for the deployment of deep-learning based tools. Self-supervised learning (SSL) is a popular technique in transfer learning, where pre-training involves completing an auxiliary task which can generate labels without human intervention (Koohbanani et al., 2021). Many simple SSL techniques proposed for natural images have been evaluated within the medical imaging community. One simple SSL task is to create a 4-way classification problem where an image is randomly rotated by 0, 90, 180, or 270 degrees and the model is tasked with correctly predicting the rotation (Gidaris et al., 2018). Another involves correctly predicting the solution to a jigsaw puzzle in which an original image is split into 9 tiles and shuffled (Noroozi and Favaro, 2016).Variants of both of these models have been shown to improve classification performance in Camelyon 16, a large histopathology dataset (Koohbanani et al., 2021).

Despite the fact that SSL can be used to extract domain-specific features from target data, models pre-trained with ImageNet, a large natural image dataset, often outperform SSL pre-training using domain-specific data. Several studies have shown that using non-medical images, including texture and natural images, as source data can improve target task performance in medical images (Li and Plataniotis, 2020; Ribeiro et al., 2017). While SSL typically uses the same dataset for pretraining (source) and fine-tuning (target), we evaluate how using source datasets from different domains (e.g. natural images, textures, and histology datasets with different tissue types and zoom levels) affects downstream performance in the classification of metastatic tissues of the Patch Camelyon (PCam) dataset.

## 2. Datasets

For the target task, we use the PCam dataset, consisting of 327,680 patches extracted from the Camelyon16 dataset (Veeling et al., 2018). Several datasets previously used for transfer learning were selected to evaluate their effectiveness as source datasets during a jigsaw SSL pre-training task: (1) *Tiny ImageNet* (C=200, $N_C$=500) (Deng et al., 2009).; (2) *Amsterdam Library of Textures (ALOT)* (C=250, N=27,500) (Burghouts and Geusebroek, 2009); (3) *Colorectal Cancer Dataset (CRC)* (C=8, N=5,000) (Kather et al., 2016). We use the shorthand C= for number of classes, $N_C$= for number of examples per class (where this is uniform) and N= for number of examples in the dataset.

## 3. Model Implementation

All SSL pre-training and fine-tuning tasks used the ResNet-34 model. To train all models we used stochastic gradient descent with the Adam optimizer, cross-entropy loss, momentum of 0.9, learning rate of 0.001, and weight decay of 0.001 for 100 epochs. For SSL pre-training the batch size used was 256 and for fine-tuning 64.

**Rotation SSL pre-training:** 4,000 images from each source dataset were randomly rotated and the model was tasked with predicting rotation class (Gidaris et al., 2018).

**Jigsaw pre-training:** 4,000 images from each source dataset were divided into 9 evenly sized tiles and were shuffled according to 100 pre-determined jigsaw patterns. Each tile was forwarded through the model and the outputs were concatenated according to the order specified in a randomly selected jigsaw solution (Noroozi and Favaro, 2016).

**Binary metastatic tissue classification:** A model was trained and evaluated using a reduced PCam dataset in 2 configs: $N_C$=1,000 (0.76%) and $N_C$=100 (0.076%). SSL pre-trained networks were compared to models trained either from scratch (Random) or ImageNet.

**Diversity evaluation:** We implemented a diversity metric used by DeVries et al. (2020) where images from each source dataset were embedded into a pre-trained feature embedding. We fit a Gaussian distribution to the embeddings and computed the average likelihood.

## 4. Results

Performance of jigsaw SSL pre-trained models for all source datasets was comparable or exceeded networks pre-trained with ImageNet and with random initializations (Table 1). This was particularly apparent in the lowest data regime ($N_C$=100). Conversely, in the lowest data regime for rotation SSL pre-training, ImageNet pre-training yielded the best performance.

We found a weak correlation between diversity rank and model performance in the N=100 condition, while the $N_C$=1,000 condition was uncorrelated. The CRC dataset which exhibited the highest diversity was found to be a top ranking initialization for both data regimes in Jigsaw SSL and $N_C$=100 Rotation SSL, outperforming pre-training using PCam.

## 5. Conclusions

This study suggests that using datasets from different domains can yield comparable or superior initializations to homogeneous pre-training. While heterogeneous pre-training did not

Table 1: Classification performance on PCam dataset across different initializations

| Initialization | Likelihood (Div. Rank) | Jigsaw Accuracy (%) (Rank) | | Rotation Accuracy (%) (Rank) | |
|---|---|---|---|---|---|
| | | $N_c$=100 | $N_c$=1,000 | $N_c$=100 | $N_c$=1,000 |
| ImageNet | - | 75.5 ± 2.1 | 83.3 ± 0.9 | 75.5 ± 2.1 | 83.3 ± 0.9 |
| Random | - | 74.1 ± 1.0 | 77.9 ± 0.7 | 74.1 ± 1.0 | 77.9 ± 0.7 |
| SSL PCAM | 9850.05 (3) | 74.2 ± 4.9 (4) | 83.8 ± 0.3 (3) | 67.8 ± 2.1 (3) | 83.3 ± 0.9 (2) |
| SSL CRC | 9848.12 (1) | 76.8 ± 3.9 (1) | 84.1 ± 0.7 (1) | 74.2 ± 3.6 (2) | 82.5 ± 0.7 (4) |
| SSL TinyImgNet | 9849.78 (2) | 75.2 ± 6.3 (3) | 83.8 ± 1.7 (2) | 75.0 ± 2.2 (1) | 82.8 ± 1.0 (3) |
| SSL ALOT | 9850.48 (4) | 76.2 ± 6.3 (2) | 83.7 ± 1.0 (4) | 67.3 ± 5.0 (4) | 83.6 ± 0.7 (1) |

show a statistically significant improvement due to the small number of runs, we observed that including more diverse data outside of the target dataset can improve performance for some SSL tasks in low-data conditions.

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
