# OpenReview forum: "Strength in Diversity: Understanding the impacts of diverse training sets in self-supervised pre-training for histology images"
_MIDL.io/2021/Conference/Short — MIDL 2021 Poster_

### Official Review · Reviewer_cq8x · 2021-04-19

**Confidence:** 4
**Final Rating:** 3

**Summary:**

The paper attempts to examine the relationship between the diversity of the source data with the performance of the downstream classification task. The experiments examine two SSL approaches with 100 and 1000 training data. The paper claims to identify a weak correlation between diversity and target dataset performance.

**Strengths:**

1. The relationship between task diversity, especially out-of-distribution, and the target performance is an important topic of research.
2. The evaluation at the low-data regime is interesting. The imagenet and random baselines are informative.

**Weaknesses:**

1. The differences in likelihood between datasets are rather small and make it difficult to fully understand the significance of relative diversity ranks.
2. Many results in Table 1 are not statistically significant with overlapping confidence intervals, which makes the conclusions rather weak.

**Deanonymize Review:**

no

**Detailed Comments:**

Instead of ranking the diversity of different datasets, one alternative to increasing the diversity is to combine different datasets together.

What do "homogeneous pre-training" and "heterogeneous pre-training" mean exactly?

**Justification Of The Rating:**

The paper presents an interesting research direction regarding the task diversity and target dataset performance, especially when the source and target are from different distributions, but the empirical studies could be improved to better support the conclusions.

**Paper Type:**

validation/application paper

**Special Issue:**

no

---

### Official Review · Reviewer_NTFk · 2021-04-26

**Confidence:** 4
**Final Rating:** 2

**Summary:**

This work evaluates the impact of self-supervision for pre-training CNNs on source datasets that are different from the target dataset. The target task is the classification of metastatic tissue in lymph node sections (PatchCamelyon dataset). Self-supervision is applied in the form of rotation and jigsaw on multiple datasets and tasks.

**Strengths:**

The paper is interesting and well written. Self-supervision can be relevant in many domains and tasks, particularly in medical imaging. Promising preliminary results are obtained which motivate a more complete and more carefully designed analysis.


**Weaknesses:**

The motivation is weak in the introduction:
“While SSL typically uses the same dataset for pretraining… we evaluate how using source datasets from different domains ...” Can you provide motivation for this? When I reach this point, I wonder why you would want to use data from different domains with SSL. There are surely motivations for this, please state.

I think rotation as SSL does not make sense for some of your source domains. In histopathology, unlike natural images, the rotations are arbitrary. The task of classifying rotated patches into 0/90/180/270 should not be feasible. The model cannot predict how your image was rotated since tissues are scanned in a random position/direction. Same for the textures of ALOT, maybe some classes have a dominant orientation but I think most don’t. Maybe it is ok to just overfit the training set without being able to generalize to the test set, that’s already some self supervision, but I doubt this would help much the main task. In future work, I suggest carefully choosing the tasks and datasets, reporting test results on the SSL tasks, and commenting on these points. For example, a typical self-supervised task in digital pathology is to learn similarity between patches.

Model implementation:
Batch size and epochs should be specified also for the models trained from scratch/pretrained on ImageNet.
An early stopping on some validation data would be better to fairly compare the approaches. To consider for future validation of the analyses.

It would be better to compare SSL and standard pre-training on the same source datasets (or at least for some if not possible for all).


**Deanonymize Review:**

no

**Detailed Comments:**

Maybe use different notations to differentiate between source and target dataset numbers (C, Nc and N)


**Justification Of The Rating:**

The problem tackled in this paper is interesting and it could be an interesting preliminary result. However, I found major weaknesses as reported above, particularly in the choice and motivation of SSL tasks and methods comparisons.

**Paper Type:**

validation/application paper

**Special Issue:**

no

---

### Meta-Review · Program_Chairs · 2021-05-11

**Recommendation:** Accept (Poster)
**Confidence:** 5

**Metareview:**

Two reviewers have different opinions about this work. The main concern from review is the motivation to use SSL. However, the proposed method could be interesting in this domain and may inspire other tasks. The proposed model is also evaluted with suffient experiments. I therefore recommend acceptance of this paper.

---

### Decision · Program_Chairs · 2021-05-11

Accept (Poster)